# Rheological Characteristics of Soluble Fibres during Chemically Simulated Digestion and Their Suitability for Gastroparesis Patients

**DOI:** 10.3390/nu12082479

**Published:** 2020-08-17

**Authors:** Harsha Suresh, Vincent Ho, Jerry Zhou

**Affiliations:** 1School of Medicine, Western Sydney University, Campbelltown NSW 2560, Australia; 17271790@student.westernsydney.edu.au (H.S.); v.ho@westernsydney.edu.au (V.H.); 2Gastrointestinal Motility Disorders Unit, Western Sydney University, Campbelltown NSW 2560, Australia; 3University Medical Clinic of Camden & Campbelltown (UMCCC), Campbelltown NSW 2560, Australia

**Keywords:** dietary fibres, soluble fibre, gastroparesis, motility disorder, rheology

## Abstract

Dietary fibres are an integral part of a balanced diet. Consumption of a high-fibre diet confers many physiological and metabolic benefits. However, fibre is generally avoided by individuals with gastrointestinal motility disorders like gastroparesis due to increased likelihood of exacerbated symptoms. Low-viscosity soluble fibres have been identified as a possible source of fibre tolerable for these individuals. The aim of this study is to determine the rheological properties of 10 common commercially available soluble fibres in chemically simulated digestive conditions and evaluate their suitability for individuals with mild to moderate gastroparesis, a gastric motility disorder. Rheological testing under neutral condition (distilled water pH 7) and chemically simulated gastric digestion were evaluated to determine the yield point and relative viscosity of each fibre. Our results reveal two rheological categories of soluble fibres; pseudoplastic and dilatant. Simulated digestion was shown to significantly alter the yield-points of psyllium husk, iota-carrageenan, beta-glucan, apple-fibre pectin, and inulin. Gum Arabic and partially hydrolysed guar gum showed the lowest viscosities and were not affected under simulated digestion, characteristics that make them potential candidate fibres for patients with gastroparesis. Altogether, our results demonstrate that digestion can have a significant impact on fibre viscosity and should be taken into consideration when evaluating the suitability of fibres for patients with gastric motility disorders.

## 1. Introduction

Dietary fibres are an integral part of a balanced diet. An adequate daily intake of fibre is essential to maintain good gut health, reduce the risk of diabetes, heart disease and colorectal cancer [1,2,3,4]. Dietary fibres are defined by Food Standards Australia New Zealand (FSANZ) as “The fraction of edible parts of plants or their extracts, or synthetic analogues, that are resistant to digestion and absorption in the small intestine, usually with complete or partial fermentation in the large intestine. This includes polysaccharides, oligosaccharides (degree of polymerisation >2) and lignans” [5]. Such a broad definition of dietary fibres indicates that there are many different types with attendant structural chemistry, rheological properties and physiological effects. Physicians recommend that healthy adults should consume 25–30 g of fibre daily as part of their balanced diet, but most adults and children fail to meet their daily intake requirement [6,7]. Shortfalls in dietary fibre intake can lead to poor gut health, increased risk of obesity and chronic gastrointestinal (GI) disorders such as constipation, diarrhoea and IBS [8].

There are primarily two types of fibre in food—soluble and insoluble. Both these types of dietary fibre are indigestible in the GI tract but play an important role in gut health and motility. Soluble fibres become ‘sticky’ and absorb water in the digestive system to form gel-like substances that help modulate blood glucose, reduce cholesterol and reduce gastroesophageal reflux disease symptoms [9,10,11]. Soluble fibres play a role in improving lower GI health through fermentation by gut bacteria and the release of beneficial by-products, including short chain fatty acid (SCFA) acetates, propionates and butyrates [12,13]. Soluble fibres such as psyllium, β-glucan, pectin and guar gum form colloidal gels and hydrocolloid pastes that are ideally suited to provide cholesterol reduction, glycaemic control and early satiety in patients with type 2 diabetes mellitus, laxation for patients with constipation and irritable bowel syndrome and colonic relief in patients with diarrhoea [14,15]. The addition of dietary fibres has been shown to affect the viscosity of small intestinal digesta and the bioavailability of antioxidants [16,17].

Gastroparesis is a GI motility disorder where there is reduced gastric functionality and varying degrees of stomach paralysis. Sufferers experience a range of post-prandial symptoms associated with delayed gastric emptying [18]. Despite the benefits of soluble fibres, they are not recommended to patients with gastroparesis due to the risk of exacerbated symptoms such as bloating, nausea, abdominal pains and vomiting [18,19,20,21,22]. Certain soluble fibres are shown to cause delayed gastric emptying, and this has been verified using ^99m^Tc-sulphur colloid radio-labelled meals and gastric-emptying scintigraphy [23]. Gastroparesis patients also suffer from 30–40% co-morbidity with either type 1 or 2 diabetes mellitus where soluble dietary fibres can play an important part in glucose modulation [24]. A long-term lack of fibre can lead to dysbiosis and poor gut health; therefore, viable soluble dietary fibre options are needed for gastroparesis sufferers [21,24].

Properties such as the molecular weight (MW), chain length, chemical structure and branching, the preparation solvent, pH and the thermodynamic configuration of constituent polysaccharides in solution play a significant role in determining the rheological properties of a dietary fibre preparation [25,26,27]. A category of soluble fibre called “low-viscosity fibres” have recently gained significant interest due to the ease in which they can be emptied through the GI tract, while displaying significant metabolic and physiological benefits. Dilution and hydrolysis of dietary fibres are often employed commercially in order to decrease the viscosity of food preparations, a prominent example of this being partially hydrolysed guar gum (PHGG) soluble fibre, which is manufactured by the hydrolysis of guar gum soluble fibre. The beneficial effects of low-viscosity type soluble dietary fibres have been documented in the literature [28,29,30]. Low-viscosity fibres have been used to alter the texture, rheology, taste and colour of manufactured food products to assist in glucose modulation, body weight management and as prebiotic supplements.

The sum of existing literature indicates a need to characterise the rheological behaviour of certain commonly available soluble dietary fibres in depth. Such characterisations would greatly aid healthcare providers in determining whether certain soluble dietary fibres can be incorporated into foods designed for gastroparesis patients. It has been noted in the literature that early satiety, a symptom of gastroparesis, is better correlated with the viscosity of gastric-phase digestion rather than the viscosity of intestinal-phase digestion [31,32]. Hence, the rheological properties of fibres under simulated upper GI digestion would provide a more accurate assessment of fibre suitability for gastroparesis patients [33].

The primary aim of this study is to determine the rheological properties of 10 soluble fibres during chemically simulated upper GI digestion. Chemical digestion is performed in order to identify candidate fibres suitable for patients with mild-to-moderate gastroparesis symptoms who are capable of oral feeding. By examining the rheological behaviour of the selected fibres at varying concentrations and conditions, the authors hypothesise that “low-viscosity” soluble fibres would retain low yield-point shear stress values at crossovers during simulated digestion.

## 2. Materials and Methods

### 2.1. Instrumentation and Equipment

The primary instrument was the Dynamic Stress Rheometer (DSR)™ and CPU from Rheometric Scientific, Texas Instruments (Piscataway, NJ, USA) which included a Neslab RTE11 water bath and fibre Dry™ air-dryer from Pisco (Elmhurst, IL, USA). A 40.0-mm-diameter, smooth, chrome-finished aluminium bottom plate; 40.0-mm-diameter, smooth, hard-coated aluminium upper plate; and the hard-coated aluminium adapter for the upper plate used in the analysis were all acquired from Rheometric Scientific, Texas Instruments (Piscataway, NJ, USA).

### 2.2. Chemicals and Reagents

The dietary fibre samples were purchased from various commercial suppliers in Australia and the labelled dietary information on the products is shown in Table 1. The following chemical standards with reported analytical purity were procured from multiple suppliers. Sodium chloride (99.7%), potassium chloride (99.0%), sodium bicarbonate (99.7%) and hydrochloric acid (32.0%) were purchased from Chem-Supply (Gillman, SA, Australia). The pepsin (99.0%) reference standard was procured from European Pharmacopoeia (Strasbourg, France) and α-amylase (99.0%) was purchased from Sigma-Aldrich (St Louis, MO, USA). The purified de-ionized water used in the analyses (>18 MΩ.cm) was obtained from a MilliQ™ Advantage A10 system with a Q-POD from Merck (Darmstadt, Germany).

### 2.3. Experimental Procedure

The simulated salivary fluid (SSF) and simulated gastric fluids (SGF) were prepared in accordance with the Davis and Minekus methods reported in the literature [34,35]. Distilled water (pH 7) was collected in a 1-L Schott bottle from de-ionized water. The SSF was prepared by accurately weighing out potassium chloride (0.4 mM), sodium chloride (0.4 mM), sodium bicarbonate (5.0 mM) and α-amylase (2.0% *w*/*v*) into a 250-mL Schott bottle and made up to volume with 200 mL of de-ionized water. The SGF solutions were prepared by accurately weighing out sodium chloride (34.2 mM) and pepsin (0.0525% *w*/*v*) into a 1-L Schott bottle. The SGF solution was adjusted to physiological gastric pH 2 and made to volume with 1 L of de-ionized water. An additional SGF solution was made to pH 4 to simulate gastric condition of individuals on proton-pump inhibitors (PPI). It has been reported that approximately 60% of gastroparesis patients also suffer gastro-oesophageal reflux disorder [24] and are therefore prescribed PPI, which reduces gastric acid production and elevates gastric pH from 2 to 4 [36]. The SGF at pH 4 was used to accurately represent the physiological environment within these patients. The SGF solutions (pH 4 and pH 2) were created by the dropwise addition of dilute hydrochloric acid (6.4% *v*/*v*) and pH adjustment using a calibrated pH meter (Mettler Toledo, Port Melbourne, VIC, Australia). All the prepared solutions were then sonicated at 60 °C for 30 min and then cooled for 20 min. The sonication and cooling procedures were performed each time the solutions were used for analysis. The solutions were re-sealed and stored in a storage cabinet after each use with a 1-week expiry. Before analysis under the SDF-PPI (simulated digestion fluid proton pump inhibitor at pH 4) and SDF (simulated digestion fluid pH 2) conditions, the SSF and SGF solutions were sealed and placed in a water bath at 37 °C. The experimental rubric for the preparation of the dietary fibre samples is shown in Figure 1. Under each condition, rheological measurements were taken at 30 min after sample preparation. The sample beakers were sealed with aluminium foil and stored in a fume hood and then re-opened for rheological measurements at 60 min.

### 2.4. Rheological Method

The data acquisition and analysis were performed using the RSI Orchestrator v6.5.8 software from Rheometric Scientific (Piscataway, NJ, USA). The sample geometry was initialized using the stored geometry “[Para Plate] 40 mm dia PP Geometry P0019”. The smooth parallel-plate geometry was selected for the rheological analysis since a large gap of 0.3–1.0 mm can be used to reduce shear during loading and axial stresses during oscillation, which is proven effective for larger particle-size colloidal gels and hydrocolloid pastes that are formed by dietary fibre preparations [37,38,39]. The plate diameter was set at 40.0 mm and the gap between the smooth parallel plates was set at 1.0 mm and auto calibrated. The minimum sample volume was 1.257 cm^3^ and the tool serial number was 0019. The pre-defined “[DStresSwp] Dynamic Stress Sweep Test” test setup was used in the method.

### 2.5. Data Acquisition and Analysis

The dynamic stress sweep test was stress-controlled with a frequency of oscillation (ω) of 45.0 rad/s (7.28 Hz) and the sweep mode was linear. The rotation of the smooth parallel plates created the flow behaviour needed to measure the viscoelastic properties of the sample and the oscillation enables destruction free, highly precise movements that are used to measure within the sample’s viscoelastic range. The lower plate temperature was set at 37 °C to match standard human biological temperature. The samples were analysed rapidly (1–2 min) to ensure that they did not dry out on the plate, leading to consistent measurement across all sample matrices. The measurement concentration, sample state at measured concentration, rheological behaviour type, shear stress increment, initial applied shear stress (min: 0.078 Pa), final applied shear stress (max: 3901.942 Pa) and measurement time period for each dietary fibre are shown in Table 2. The target sample concentration for each dietary fibre preparation was gradually increased from 50 to 1000 mg/mL until a yield-point crossover was observed. For the sake of analysis, measurements were taken at 50, 200 and 1000 mg/mL concentration thresholds to compare yield points among the selected soluble dietary fibres.

The collected rheological measurements were statistically fitted by the application of Hook’s law for viscoelastic materials [40]. The thixotropy of the soluble dietary fibres was determined by the application of the Cox-Merz rule, where the functional dependence of complex viscosity (η*) magnitude is expressed as a function of frequency (ω) which is identical to functional dependence of the steady shear viscosity (η) which is expressed as a function of shear rate (γ) [41]. The equation for the Cox-Merz rule [42] reads as:|η*(ω)| = η(γ) | γ = ω

After data acquisition, G’ (the elastic modulus responsible for energy storage in the sample), G” (the viscous modulus responsible for energy loss in the sample) and Tan (δ) (the loss or damping factor at phase angle δ) were logarithmically plotted using the in-built RSI orchestrator software. At the initial applied shear stress, the behaviour of G’ and G” was linear and these moduli deformed to produce the G^c^ (the crossover modulus) as oscillatory shear stress was gradually increased over time. The yield-point shear stress (τ_y_) and G^c^ at the phase transition (or sol-gel transition) point were interpolated in each plot for each sample where:G’ = G” | Tan (δ) = 1

The complex viscosity (η*) was also plotted against increasing oscillatory shear stress to determine the type of rheological behaviour. The yield points and G^c^ values were tabulated and analysed using Microsoft Excel (Office 2016). The *p*-values reported in the results and discussion section were generated using a two-tailed, homoscedastic (two sample equal variance) Student’s *t*-test.

## 3. Results

### 3.1. Rheological Plots

Rheological analysis of the soluble dietary fibres produced two distinct types of thixotropic phase transitions, pseudoplastic (shear-thinning) and dilatant (shear-thickening), which occur in non-Newtonian type viscoelastic gels and pastes. Figure 2 and Figure 3 show examples of each type of phase transition with arrows indicating the yield point. In both these figures, the elastic G’ modulus, the viscous G” modulus, the complex viscosity (η*) and Tan (δ) are tracked as the shear stress is increased. The majority of soluble fibres took the form of a pseudoplastic (shear-thinning) phase transition shown in Figure 2 at the sample concentration when the yield point was achieved, where initially G’ is greater than G”. As greater oscillatory shear stress (τ) is applied to the sample, G’ gradually decreases, and after the yield-point crossover, G” is greater. Once the yield point is crossed, the complex viscosity (η*) decreases rapidly as shown in Figure 2, indicating a complete breakdown in the molecular associations of the constituent dietary polysaccharides and therefore the sample begins to the “flow” from this point onward with G” dominating the rheological behaviour.

For PHGG and gum Arabic at 50 mg/mL, G” is greater than G’ with no yield point observed in the acquisition range for shear stress (0.078–3901.942 Pa). This type of viscoelastic behaviour at low concentration is almost entirely viscous and Newtonian, since the fibre is completely dissolved in solution. This viscous Newtonian behaviour at 50 mg/mL was also found in the dietary fibres which produced pseudoplastic yield-point crossovers at 200 and 1000 mg/mL. As concentrations of PHGG and gum Arabic preparations were increased to 1000 mg/mL phase transitions and yield points were observed as shown in Figure 3. As the shear stress (τ) was increased, the complex viscosity (η*) of the sample increased until a steady-state was reached, indicating complex shear behaviour that is initially dilatant (or shear thickening). The type of behaviour where the G’ contribution is higher than the G” contribution is a non-Newtonian, thixotropic characteristic of gum Arabic and PHGG in the distilled water (pH 7), which is consistent with previous rheological studies of these fibres [43]. The dilatant rheological behaviour of gum Arabic and PHGG remained relatively consistent during simulated digestion.

### 3.2. Rheological Yield Points

The tabulated summary of the yield-point shear stress (τ_y_) and crossover modulus (G^c^) values under distilled water and simulated digestion are shown in Table 3. The sample measurements were acquired from sequential samples prepared in triplicate at both 30 and 60 min. The percentage relative standard deviation (%RSD) values across all measurements for yield shear stress measurements does not exceed ±12.81% (Citrus pectin in SDF-PPI). The %RSD values across all sample measurements for crossover modulus (G^c^) does not exceed ±11.80% (Apple-fibre pectin in distilled water). The low %RSD values for both yield-point shear stress and G^c^ in the measurements across all dietary fibre preparations indicate good method precision, repeatability, and reliability [37].

Among the 10 selected fibres in this study, preparations with yield points at 50 mg/mL are labelled as ‘high-viscosity’, preparations with yield points at 200 mg/mL are labelled as ‘medium-viscosity’ and preparations with yield points at 1000 mg/mL are labelled as ‘low-viscosity’. Such descriptions are relative terms with respect to the possible clinical use and does not describe the range of distinct rheological properties and behaviours, which is largely dependent on the preparation concentration and chemical composition of a dietary fibre supplement.

For the soluble dietary fibre preparations measured at 50 mg/mL, as shown in Figure 4, under distilled water conditions, guar gum exhibits the highest yield-point shear stress (994.51 Pa) and psyllium husk exhibits the lowest (56.31 Pa). At 30 min, no significant differences in shear stress yield points were observed between simulated digestions (pH 4) and in distilled water (*p* > 0.05), apart from iota-carrageenan (*p* = 0.03). Under simulated digestion (pH 2), psyllium husk (80.85 Pa) and iota-carrageenan (1232.03 Pa) display significantly increased yield stress (*p* < 0.01) along with xanthan gum (*p* < 0.05), while no changes were observed in guar gum. At 60 min for both simulated digestive conditions, changes in yield stress were observed for psyllium husk and iota-carrageenan (*p* < 0.05) while no major changes were observed for xanthan gum and guar gum (*p* > 0.05).

When the sample concentration was increased to 200 mg/mL as shown in Figure 5, yield points were observed for citrus pectin, apple-fibre pectin and beta-glucan. Under distilled water (pH 7), citrus pectin exhibited the highest yield-point shear stress (3049.43 Pa) and apple-fibre pectin exhibited the lowest yield shear stress (23.27 Pa). At 30 min under simulated digestive conditions, citrus pectin did not show any major changes in yield-point stress (*p* > 0.05), while apple-fibre pectin showed a minor decrease in yield-point shear stress in simulated digestion at pH 4 (*p* = 0.03). Beta-glucan showed a stepwise decrease in yield-point stress from distilled water (pH 7) to simulated digestion at pH 4 (*p* < 0.05) and then pH 2 (*p* = 0.01). At 60 min, citrus pectin exhibited no significant changes in yield-point stress (*p* > 0.05) while beta-glucan and apple-fibre pectin exhibited a significant increase in yield-point stress compared to the previous time point (*p* < 0.01).

At 1000 mg/mL as shown in Figure 6, yields were then observed for inulin, gum Arabic and PHGG, with pseudoplastic rheological behaviour in inulin and dilatant behaviour in gum Arabic and PHGG. In distilled water (pH 7), inulin displayed the highest yield-point shear stress (47.36 Pa) and PHGG displayed the lowest (20.01 Pa). At 30 min in simulated digestion, no significant changes in yield-point stress were observed for either gum Arabic or PHGG compared to water (*p* > 0.05). On the other hand, inulin exhibits a significant increase in yield-point stress in simulated digestions at pH 4 and pH 2 (*p* < 0.01), relative to water. At 60 min, the yield-point stress changes were not significant for PHGG and gum Arabic (*p* > 0.05), while there was a significant increase in yield-point stress for inulin in both simulated digestion solutions compared to its 30-min timepoint (*p* = 0.01).

## 4. Discussion

The rheology of 10 commercially available soluble dietary fibres were comprehensively studied under neutral (distilled water, pH 7) and simulated digestion conditions. The study identified gum Arabic and PHGG as two promising candidates for gastroparesis patients based on their low-viscosity behaviour in both distilled water and simulated digestion conditions.

In general, the rheological parameters of yield-point shear stress and complex viscosity (η*) are affected by the chemical structure, preparation concentration, pH and the presence of cations at steady biological temperature (37 °C). Due to these factors, significant variability in rheological properties were observed between our 10 soluble fibres. Fibres such as guar gum and psyllium husk required relatively high shear stresses to achieve their yield points at 50 mg/mL, suggesting that a large amount of mechanical force is required to breakdown their molecular associations. The physiological digestion of food bolus in the stomach requires the breakdown of particles to an average size of 2.0 mm, which is required for transit through the pylorus [44,45,46]. The shear stress requirement for a yield point in dietary fibre preparations can be analogous to the mechanical force needed by the stomach to churn and breakdown the molecular associations of the dietary fibre. When gastric motility is impaired or absent (as is the case in gastroparesis), fibres with high yield points may not be sufficiently broken down, which leads to delayed gastric emptying and associated symptoms [47].

It can be ascertained from Figure 4, Figure 5 and Figure 6 that the chemical structure and composition of dietary fibres play an important role in the shear stress requirements for a yield point in distilled water (pH 7). Guar gum and PHGG are good examples to demonstrate this, as PHGG is a short chain hydrolysed version of the guar gum polysaccharide, which consists of a mannose backbone and a galactose side chain (2:1). PHGG shows a far lower yield-point stress and lower viscosity than guar gum, demonstrating that shortening the length of major polysaccharides in a dietary fibre directly affects the rheological properties [48]. It must be noted that hydrolysing guar gum into PHGG also changed the sample state from a colloidal gel into a hydrocolloid paste. Such de-gelling helps lower the viscosity of PHGG and allows it to be incorporated into yogurts in order to reduce viscosity and improve texture quality [49]. Inulin is a linear short chain β-2,1 fructan polysaccharide with variable degrees of polymerisation. The alpha-inulin variety used in this study is water soluble at room temperature unlike delta-inulin, which is insoluble at temperatures below 40 °C. Inulin polysaccharides are of low molecular weight, with a range of 0.6 kDa to 7.2 kDa, leading to low yield-point stress values in distilled water (pH 7) [50].

The Gum Arabic dietary polysaccharide has a large molecular weight of 240–580 kDa but has been shown to demonstrate low-viscosity rheological behaviour in distilled water (pH 7) and in various dietary preparations [51]. This rheological behaviour is due to the extensively branched arabinogalactan polysaccharide which consists of a backbone of (1,3)-linked β-d-galactopyranosyl units, a side chain of between two to five (1,3)-linked β-d-galactopyranosyl units which is joined to the backbone with (1,6)-linkages and monomer residues of rhamnose, arabinose, glucuronic acid. The proportion of the amphiphilic micellar structured arabinogalactan protein (AGP) complex in gum Arabic has been shown to be correlated with increased shear-thinning thixotropy and a reduction in viscosity [52]. This effect is due to AGP micelles in solution being disrupted by the application of steady shear stress. When the steady shear stress is ceased, the AGP polysaccharide returns to its original micelle configuration. Interestingly, significantly different yield points were observed in pectin fibres from apple and citrus (Figure 5). The discrepancy may be due to varying proportions of the constituent pectin polysaccharides homogalacturonan, xylogalacturonan, apiogalacturonan rhamnogalacturonan I and rhamnogalacturonan II [53], as well as differences in the total dietary fibre content (apple fibre 40% vs citrus fibre 55%) in their respective supplements, as reported in Table 1.

It has been documented in the literature that the rheological properties of some water-soluble polysaccharides (such as, pectins, gums, mucilages) are affected by the adjustment of pH, the presence of cations and temperature [54]. In this study, the presence of H^+^, Na^+^ and K^+^ cations within the simulated gastric fluids are the primary factors that affect molecular associations of the fibre polysaccharides, but this effect is greatly dependent on whether the cations present in solution can disrupt and rearrange the default thermodynamic configuration of a polysaccharide [55]. In the polysaccharides of gum fibres (gum Arabic, PHGG, guar gum, xanthan gum), which lowered pH and increased cations, seems to have little to no effect on the yield-point shear stress requirements or the sample viscosity due to the hydrogen and covalent bonding in micellar configurations being stable and undisrupted [56,57]. The pectin-type dietary fibres (apple-fibre and citrus pectin) were similarly unaffected by the simulated gastric conditions, but these pectin dietary fibres did not form the colloidal gels reported in the “egg-box” model [58,59] and instead formed hydrocolloidal pastes, which may have affected their rheology [58]. The rheological properties of high-methoxy pectins (like apple-fibre and citrus pectin) are largely unaffected by the decrease in pH and increased cation presence and are shown in the literature to be more affected by substances like the food additive gelatine [53]. It must also be noted that the solutions used in our experiments contained little to no free Ca^2+^ ions, which are required for pectin to form the colloidal gels found in the “egg-box” model where the Ca^2+^ ions interact with COO^−^ ions in-between sliding sheets of polysaccharide chains, which then stack over each other and result in increased viscosity.

Fibres such as iota-carrageenan, inulin, beta-glucan and psyllium husk display significant changes in yield-point shear stress in simulated digestion conditions compared with distilled water. For the iota-carrageenan polysaccharide, the mechanism of gel formation and solubility is primarily affected by the presence of the OSO_3_^−^ ester sulphate group on its O-3-substituted β-d-galactopyranosyl and O-4-substituted 3,6-anhydro-α-d-galactopyranosyl dimer backbone with a double helical configuration [60]. The iota-carrageenan variety is more sulphated than kappa-carrageenan, forming a soft gel rather than the rigid gel found in kappa-carrageenan [61]. Abundance of free cations in solution results in cationic interaction where the sulphate ester groups on the crosslinked double helix configuration are aggregated, especially by K^+^. The sol-gel transition points after such interactions have been studied in the literature using the photon transmission technique [62] and are shown to increase gel formation, decrease solubility and significantly increase yield-point shear stress. Similarly, inulin also exhibited increased yield-point shear stress due to its six-turn helical configuration of the alpha-inulin polysaccharide in aqueous solution, which is affected by the aggregation of cations to the helix in a manner very similar to iota-carrageenan [63,64,65]. Although inulin has relatively low viscosity, the significant increase in viscosity under simulated digestion is not ideal for gastroparesis patients. Given the delayed gastric emptying characteristics of gastroparesis, inulin may be exposed to gastric fluids for over 4 h in these patients, resulting in increased viscosity and likelihood of associated symptoms. In addition, inulin can cause severe reactions in people with fructan allergies and is, therefore, excluded from any potential clinical study involving gastroparesis patients [66,67].

Psyllium husk is an anionic mucilage polysaccharide with many COO^−^ groups and a great deal of electrostatic repulsion, which causes the polysaccharide strands to expand and become interpenetrated. The microstructure of psyllium husk is extensively porous in distilled water and in alkaline conditions (pH > 7). When the pH is sufficiently decreased, the net intermolecular electrostatic repulsion in the polysaccharide is reduced causing the colloidal material to become more rigid, increasing the yield-point shear stress and increasing viscosity [68]. Such intermolecular associations explain its increased yield-point stress at low pH solutions. Beta-glucan is a water-soluble mucilage polysaccharide with repeating β-D-glucose monomer units with (1,3) and (1,4) glycosidic linkages that can be branched [69]. Literature investigations of beta-glucan report that the rheological viscosity is increased under acidic conditions and decreased under alkaline conditions [70]. This runs counter to the result obtained in this study, where the yield-point shear stress values and viscosity after yield are reduced more when pH is lowered in simulated digestion conditions. Such an effect may be due to the presence of cations in solution since their effect on beta-glucan has not been investigated extensively. The reduced viscosity may also be due to the low percentage of total dietary fibre present in beta-glucan (44%) and other constituents within the supplement. The other polysaccharides present in the beta-glucan supplement (22%) and their intermolecular interactions with the beta-glucan (22%) polysaccharide in aqueous solution may also have affected the rheology when the pH is reduced.

Simulated digestions were carried out over the course of 60 min, iota-carrageenan, psyllium husk, beta-glucan, apple-fibre pectin showed significant increases in yield-point shear stress compared to their 30 min timepoint results. The primary factors playing a role in yield points which increase at 60 min are the water-binding ability of constituent polysaccharides and the syneresis effect. The binding of water by dietary fibres can be done through anionic interactions, ionic interactions involving carboxyl groups with cations, strong and weak types of hydrogen bonding, hydrophobic interactions involving the formation of water clathrates on gel surfaces and the enclosure of water through capillary action [71]. Some soluble dietary fibres experience rapid syneresis during the stabilization or rest period where the sample matrix expels water bound to the constituent polysaccharides. This causes colloidal gels and hydrocolloid pastes to become brittle and dried out, increasing the yield-point shear stress and the viscosity, though this process is reversible [72]. Therefore, the water-binding capacity of soluble dietary fibres and any observed syneresis during rheological measurements are largely dependent on the chemical structure and thermodynamic configuration of constituent polysaccharides in aqueous solution.

There are some limitations in this study. While the rheological properties of the dietary fibres under neutral and simulated digestive conditions were studied, the direct physiological effects of these fibres in the upper gastrointestinal tract have not been evaluated. Though the yield-point shear stress values provide valuable rheological information for pre-clinical evaluation, they may not be an accurate representation of the peristaltic forces in the stomach. Evaluation of the physiological effects may require either a simulated stomach like a SIMulator Gastro-Intestinal (SIMGI) compartment [73] or clinical study involving human participants. Despite these limitations, the rheological data presented in this research can prove useful for clinicians and dietitians in assessing the suitability of certain dietary fibre supplements for patients with upper gastrointestinal disorders such as gastroparesis where peristaltic activity is greatly reduced or impaired.

## 5. Conclusions

Altogether, our results demonstrate the rheological heterogeneity within a range of soluble fibres. We propose that digestion may have a significant impact on fibre viscosity and should be taken into consideration when evaluating the suitability of fibres for patients with gastric motility disorders. Although the physiological benefits of soluble fibres in gastroparesis will need to be evaluated, our results provide evidence for gum Arabic and PHGG to be tolerable sources of soluble fibre for patients with gastroparesis. Future clinical studies would involve introducing our candidate fibres (gum Arabic and PHGG) into the diets of gastroparesis patients in order to evaluate their effects on blood glucose, gastric motility and associated gastric symptoms.

## Figures and Tables

**Figure 1 nutrients-12-02479-f001:**
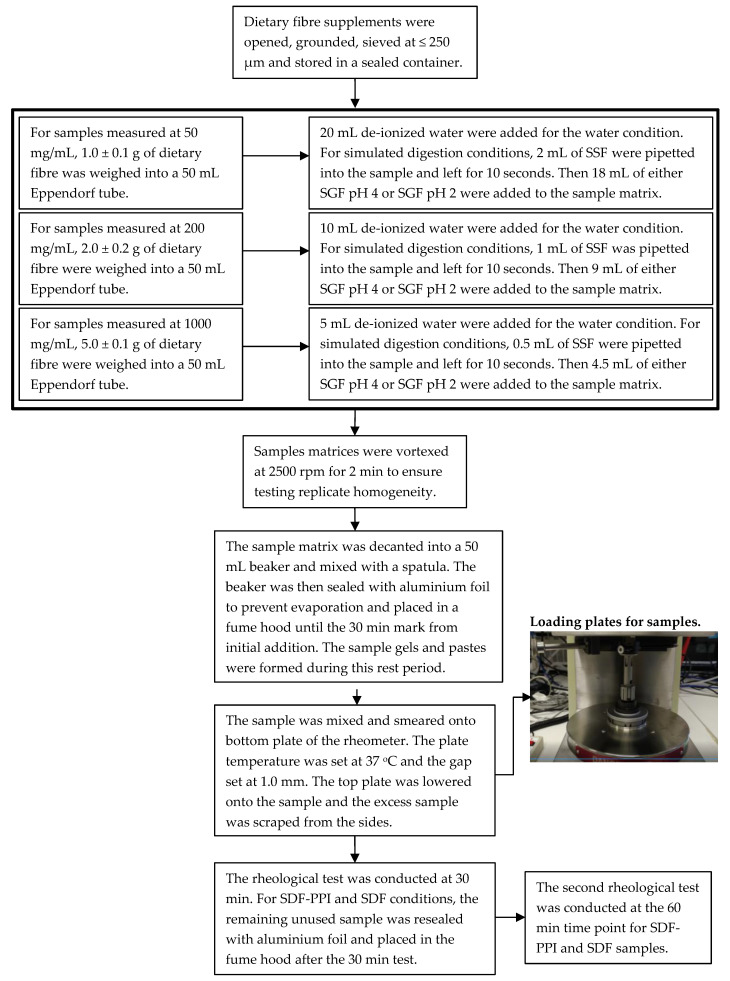
Procedural flowchart for sample preparation and rheological analysis. Keywords as follows: SSF (simulated salivary fluid), SGF (simulated gastric fluid), SDF-PPI (simulated digestion fluid proton pump inhibitor at pH 4), SDF (simulated digestion fluid at pH 2).

**Figure 2 nutrients-12-02479-f002:**
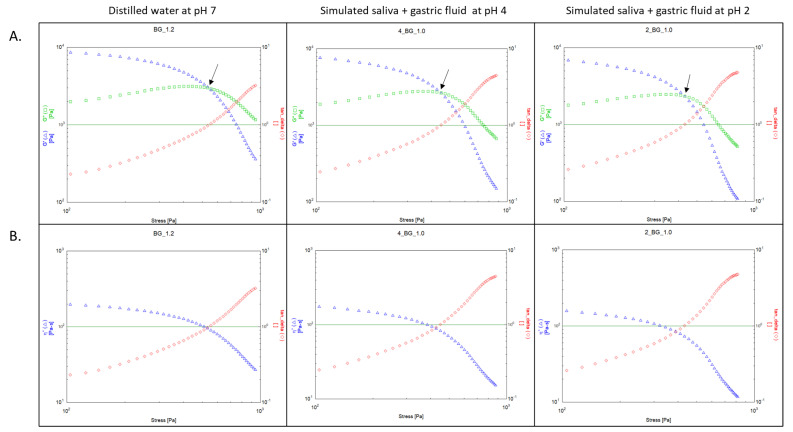
An example of pseudoplastic rheological behaviour observed in beta-glucan. Graphs show (**A**) rheological yield points (top half, y-axis left: G’ and G” in Pa, y-axis right: Tan (δ), x-axis bottom: shear stress in Pa) and (**B**) complex viscosity (bottom half, y-axis left: η* in Pa.s, y-axis right: Tan (δ), x-axis bottom: shear stress in Pa) plots as shear stress is gradually increased in distilled water (pH 7), simulated digestions with SDF-PPI (simulated digestion fluid proton pump inhibitor at pH 4) and SDF (simulated digestion fluid at pH 2). The black arrow in each plot indicates the yield point where Tan (δ) = 1. Note that the rheological behaviour does not change under chemically simulated digestion.

**Figure 3 nutrients-12-02479-f003:**
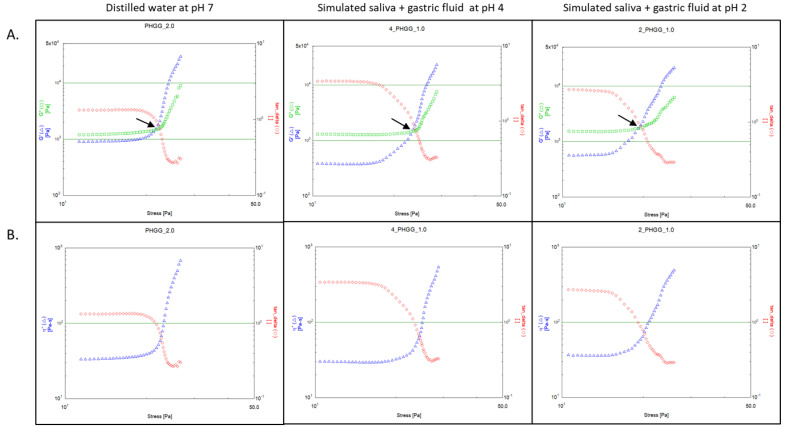
An example of dilatant rheological behaviour observed in partially hydrolysed guar gum (PHGG). Graphs show (**A**) rheological yield points (top half, y-axis left: G’ and G” in Pa, y-axis right: Tan (δ), x-axis bottom: shear stress in Pa) and (**B**) complex viscosity (bottom half, y-axis left: η* in Pa.s, y-axis right: tan (δ), x-axis bottom: shear stress in Pa) plots as shear stress is gradually increased in distilled water (pH 7), simulated digestions with SDF-PPI (simulated digestion fluid proton pump inhibitor at pH 4) and SDF (simulated digestion fluid at pH 2). The black arrow in each plot indicates the yield point where Tan (δ) = 1. Note that the rheological behaviour does not change under chemically simulated digestion.

**Figure 4 nutrients-12-02479-f004:**
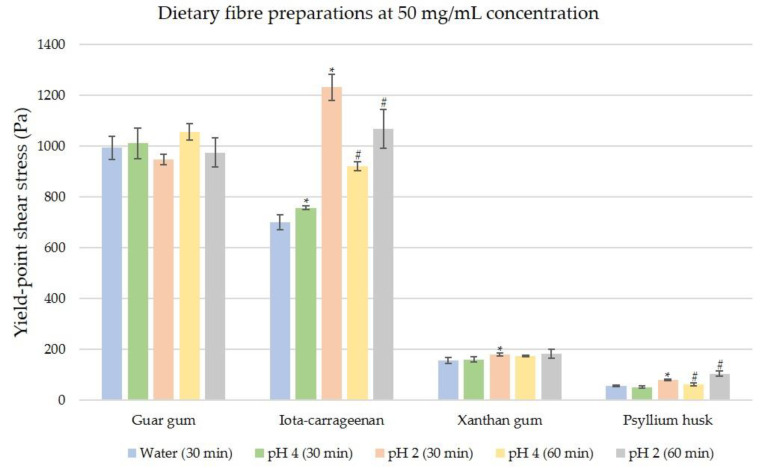
Comparisons of yield-point shear stresses of four ‘high-viscosity’ fibres; guar gum, iota-carrageenan, xanthan gum, and psyllium husk at concentration of 50 mg/mL. Test conditions are in distilled water (pH 7), and simulated digestion at pH 4 and pH 2. The data tip (*) indicates significant difference between the distilled water condition and the simulated digestion condition (pH 4 or pH 2) at 30 min. The data tip (#) indicates significant difference between the simulated digestion condition at 30 min (pH 4 or pH 2) and its corresponding simulated digestion condition at 60 min. The significance level is *p* ≤ 0.05 for both (*) and (#).

**Figure 5 nutrients-12-02479-f005:**
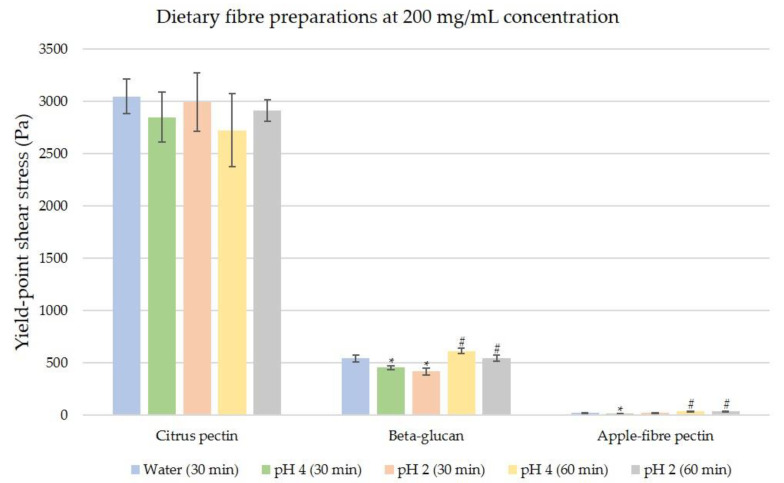
Comparisons of yield-point shear stresses of three ‘medium-viscosity’ fibres; citrus pectin, beta-glucan, and apple-fibre pectin at 200 mg/mL. Test conditions are in distilled water (pH 7), and simulated digestion at pH 4 and pH 2. The data tip (*) indicates significant difference between the distilled water condition and the simulated digestion condition (pH 4 or pH 2) at 30 min. The data tip (#) indicates significant difference between the simulated digestion condition at 30 min (pH 4 or pH 2) and its corresponding simulated digestion condition at 60 min. The significance level is *p* ≤ 0.05 for both (*) and (#).

**Figure 6 nutrients-12-02479-f006:**
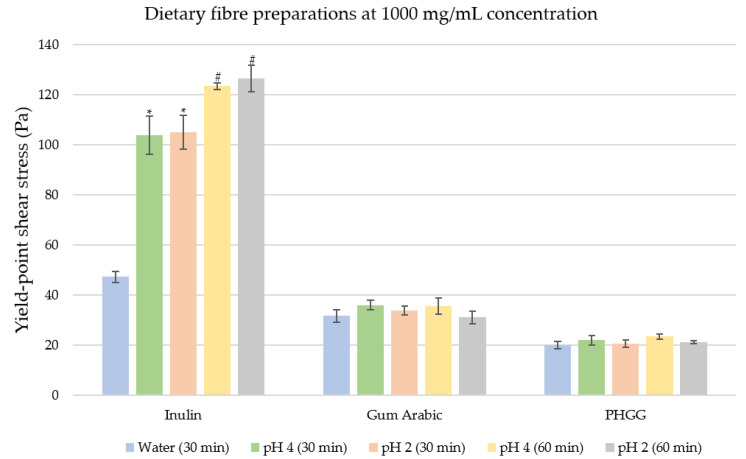
Comparisons of yield-point shear stresses of three ‘low-viscosity’ fibres; inulin, gum Arabic, and partially hydrolysed guar gum (PHGG) at 1000 mg/mL. Test conditions are in distilled water (pH 7), and simulated digestion at pH 4 and pH 2. The data tip (*) indicates significant difference between the distilled water condition and the simulated digestion condition (pH 4 or pH 2) at 30 min. The data tip (#) indicates significant difference between the simulated digestion condition at 30 min (pH 4 or pH 2) and its corresponding simulated digestion condition at 60 min. The significance level is *p* ≤ 0.05 for both (*) and (#).

**Table 1 nutrients-12-02479-t001:** Labelled nutritional content of tested dietary fibre supplements.

Product (Name) Qty (per 100 g)	Commercial Supplier (Name)	Total Dietary Fibre (%)	Energy (kJ)	Carbohydrates, Sugars (g)	Protein (g)	Total Fat, Saturated Fat (g)	Essential Nutrients (mg) *
Guar gum	Ceres Organics	77.3	743.0	0.5, 0.5	4.7	0.3, 0.0	13.0
Iota-carrageenan	The Melbourne Food Ingredient Depot	76.0	1298.0	76.0, 0.0	0.0	0.0, 0.0	640.0
Xanthan gum	Myprotein	62.0	832.0	78.0, 0.0	3.3	0.0, 0.0	0.0
Psyllium husk	SF Health Foods	80.0	802.0	0.0, 0.0	3.0	3.0, 0.0	79.0
Citrus pectin	Lotus Pantry	55.0	1005.0	30.0, 30.0	0.0	0.0, 0.0	1000.0
Beta-glucan	Blooms Health Products	44.0	1251.0	22.0, 0.0	20.0	5.0, 1.0	1088.0
Apple-fibre pectin	Myprotein	40.0	1850.0	90.0, 40.0	0.0	0.0, 0.0	0.0
Inulin	Myprotein	89.0	848.0	8.0, 8.0	0.0	0.0, 0.0	0.0
Gum Arabic	New Directions Australia	80.0	1339.0	80.0, 0.0	0.0	0.0, 0.0	0.0
Partially hydrolysed guar gum (PHGG)	Healthy Origins	80.0	1691.0	93.0, 13.0	0.0	0.0, 0.0	0.0

* Essential nutrients are the cumulative Na, Mg, Ca, K, Fe and Zn present in the commercial fibre supplement in mg.

**Table 2 nutrients-12-02479-t002:** Measured concentration, colloidal state, rheological behaviour and method analysis parameters under parallel plate configuration.

Dietary Fibre (Name)	Measurement Concentration (mg/mL)	Sample State at Measured Concentration	Rheological Behaviour at Concentration	Shear Stress Increment (Pa/s)	Initial Applied Shear Stress (Pa)	Final Applied Shear Stress (Pa)	Measurement Time Period (s)
Guar gum	50.0	Colloidal gel	Pseudoplastic	30.0	400.0	1200.0	26.6
Iota-carrageenan	50.0	Colloidal gel	Pseudoplastic	25.0	400.0	1300.0	36.0
Xanthan gum	50.0	Hydrocolloid paste	Pseudoplastic	20.0	50.0	400.0	17.5
Psyllium husk	50.0	Hydrocolloid paste	Pseudoplastic	4.0	1.0	180.0	44.8
Citrus pectin	200.0	Hydrocolloid paste	Pseudoplastic	100.0	1500.0	3900.0	24.0
Beta-glucan	200.0	Hydrocolloid paste	Pseudoplastic	20.0	100.0	900.0	40.0
Apple-fibre pectin	200.0	Hydrocolloid paste	Pseudoplastic	1.0	1.0	100.0	99.0
Inulin	1000.0	Hydrocolloid paste	Pseudoplastic	8.0	20.0	240.0	27.5
Gum Arabic	1000.0	Hydrocolloid paste	Dilatant	0.4	10.0	50.0	100.0
Partially hydrolysed guar gum (PHGG)	1000.0	Hydrocolloid paste	Dilatant	0.4	10.0	40.0	75.0

**Table 3 nutrients-12-02479-t003:** Tabulated summary of yield-point shear stresses (τ_y_), crossover moduli (G^c^)**,** and percentage relative standard deviation (%RSD) for the dietary fibre preparations in distilled water, and simulated digestion with gastric fluid at pH 4 and gastric fluid at pH 2.

Dietary Fibre (Name)	Sample Concentration (mg/mL) ^a^	Rheological Condition (pH; time point) ^b^									
		Distilled Water (pH 7; 30 min)		Simulated Digestion (pH 4; 30 min)		Simulated Digestion (pH 2; 30 min)		Simulated Digestion (pH 4; 60 min)		Simulated Digestion (pH 2; 60 min)	
		τ_y_ (Pa) ± (%RSD)	G^c^ (Pa) ± (%RSD)	τ_y_ (Pa) ± (%RSD)	G^c^ (Pa) ± (%RSD)	τ_y_ (Pa) ± (%RSD)	G^c^ (Pa) ± (%RSD)	τ_y_ (Pa) ± (%RSD)	G^c^ (Pa) ± (%RSD)	τ_y_ (Pa) ± (%RSD)	G^c^ (Pa) ± (%RSD)
Guar gum	50.0	994.51 (4.56)	1573.43 (5.46)	1011.12 (6.01)	796.77 (4.31)	946.93 (2.14)	810.08 (9.31)	1057.50 (3.02)	872.16 (7.47)	975.59 (5.91)	807.69 (7.28)
Iota-carrageenan	50.0	700.08 (4.30)	819.82 (2.18)	758.00 (0.95)	2050.50 (7.79)	1232.03 (4.16)	782.68 (8.07)	921.82 (1.85)	1480.77 (9.95)	1068.17 (7.18)	528.68 (5.28)
Xanthan gum	50.0	156.89 (7.16)	114.16 (7.48)	160.99 (6.63)	125.16 (9.63)	180.37 (3.94)	129.33 (4.41)	174.83 (1.47)	139.27 (11.63)	183.16 (9.40)	128.65 (11.64)
Psyllium husk	50.0	56.31 (6.78)	134.31 (3.12)	51.38 (9.49)	117.06 (4.04)	80.85 (4.19)	119.94 (0.88)	62.70 (9.90)	116.84 (3.11)	104.49 (10.24)	123.44 (1.66)
Citrus pectin	200.0	3049.43 (5.36)	2043.93 (6.72)	2848.50 (8.41)	2234.63 (1.99)	2996.10 (9.38)	2036.47 (6.30)	2725.07 (12.81)	1578.80 (1.97)	2911.83 (3.50)	1743.17 (8.00)
Beta-glucan	200.0	544.58 (6.08)	2938.67 (1.22)	454.57 (3.41)	2653.37 (2.88)	419.02 (7.57)	2427.70 (9.47)	614.19 (3.88)	2952.63 (1.11)	545.38 (5.28)	2784.37 (6.67)
Apple-fibre pectin	200.0	23.27 (8.49)	343.67 (11.80)	19.29 (2.53)	226.21 (11.16)	22.99 (4.88)	295.24 (5.07)	39.27 (6.56)	451.10 (9.76)	36.51 (2.13)	385.13 (10.23)
Inulin	1000.0	47.36 (4.64)	98.01 (9.52)	103.87 (7.40)	151.04 (9.14)	105.15 (6.37)	143.09 (8.68)	123.48 (0.94)	222.70 (1.07)	126.41 (4.22)	197.24 (1.03)
Gum Arabic	1000.0	31.76 (7.94)	794.02 (6.41)	36.09 (5.39)	805.88 (2.53)	33.83 (4.36)	815.66 (4.55)	35.66 (8.82)	784.39 (8.11)	31.15 (8.08)	893.63 (5.75)
Partially hydrolysed guar gum (PHGG)	1000.0	20.01 (7.52)	1460.33 (3.79)	21.99 (9.00)	1504.47 (4.17)	20.55 (8.78)	1699.07 (2.86)	23.51 (4.28)	1522.40 (3.29)	21.22 (2.30)	1904.53 (10.41)

^a^ Fibre concentration in solution was increased when no crossover modulus (G^c^) was observed in the 0.08–3901.01 Pa shear stress (τ) acquisition range; ^b^ Measurements taken at each condition with *n* = 3 replicates for yield shear stress (τ_y_) and crossover modulus (G^c^) in Pa with ± %RSD at the yield point (sol-gel transition point).

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
