# Peer review of "Rheological Characteristics of Soluble Fibres during Chemically Simulated Digestion and their Suitability for Gastroparesis Patients"

_nutrients, 2020, doi:10.3390/nu12082479_

Round 1

Reviewer 1 Report

I believe that the paper is very well explained and well written.

The only minor issue, is some redundant information, that I suggest removing as follow:

I appreciate the attention to detail but line 95 to 98 is no needed. From line 102 to 107 is no needed.

Table 1. Could the authors put the meaning of PHGG in the table?

Figure 1. the purchasing step of the fibre is no needed

Line 154 to 166 are no needed

Line 166 to line 179 are not data acquisition, they should move up to the experimental procedure

In line, 183 references are no needed.

The excel version in line 196 is no needed

Figure 3. Please adjust the axis, it is no needed that arrives to 10^2 , that way authors will not have half of the graph empty

Figure 4. Remove the .00 from the Y axis

Author Response

Reviewer 1

The authors would like to thank the reviewer for taking the time to review our manuscript and providing positive and useful feedback. We have taken the reviewer’s comments into consideration and our responses are provided below:

  1. I appreciate the attention to detail but line 95 to 98 is no needed. From line 102 to 107 is no needed.

Reply: Specifications regarding rheometer have been condensed to only describe the system’s make and model, attachments, and shear plate selection [lines 102 – 106]. The original description of the general laboratory equipment used in experiments (e.g. scale, fume hood) have been removed.    

  1. Table 1. Could the authors put the meaning of PHGG in the table?

Reply: Table 1 PHGG abbreviation has been amended to “Partially hydrolysed guar gum”.

Figure 1. the purchasing step of the fibre is no needed

Reply: Removed from Figure 1, content added to described fibre preparation for sampling (e.g. sieving and storage)

Line 154 to 166 are no needed

Reply: Description of plate options and reasoning for selection has been condensed only the plate selected (smooth parallel plate) [Line 155-158].

Line 166 to line 179 are not data acquisition, they should move up to the experimental procedure

Reply: Amended to reviewer’s suggestion. Section 2.5 “data acquisition and analysis” added [Line 162 – 176].  

In line, 183 references are not needed.

Reply: Reference for Hook’s Law removed

The excel version in line 196 is no needed.

Reply: Amended in manuscript

Figure 3. Please adjust the axis, it is no needed that arrives to 10^2, that way authors will not have half of the graph empty

Reply: Figure 3 X-axis adjusted to 50 to better represent graphs.

Figure 4. Remove the .00 from the Y axis

Reply: Amended in Figure 4

Reviewer 2 Report

In this paper, the authors present the rheological characteristics of soluble fibres during chemically-simulated digestion, with an aim to inform suitable fibre options for patients with gastroparesis. The paper is very well-written, and the authors present a comprehensive and detailed study. The following suggestions may offer opportunity for improvement.

General/major comments:

  1. The title, and paper, describes the methods as "simulated digestion". I assumed that this meant computational simulations, but in fact, the authors performed chemical simulations. I suggest expanding this for clarity. E.g., in the title, "...during chemically simulated digestion...".
  2. Intro, Line 57: The authors state, "Sufferers experience a range of post-prandial symptoms due to delayed gastric emptying." I am not aware of a defined cause-effect relationship between symptoms and gastric-emptying in gastroparesis. This statement should be referenced, although I ultimately suggest changing to the wording to reflect an association rather than a cause-effect. E.g., >>"...symptoms associated with delayed gastric emptying."
  3. Intro, Line 62: The statement on long-term health complications would benefit from including a reference(s). 
  4. Figures 2&3: How are the results different for pH 2 vs 4 vs 7? They all the same / similar to me. It would be good to mention the differences, or lack thereof, in the caption. 
  5. Figures 4, 5, & 6: Are you able to include the stats? Which of these results are statistically significant compared to the others? 
  6. Figures 4, 5, & 6: Green and grey are both used twice. It is impossible to tell which bar is which in the graph. I assume it follows the order in the legend at the bottom, but that is just an assumption. Different colours should be used for clarify.  
  7. The paper is extremely well written, but it is also very long. Each of the sections are lengthy, and at some points, the length is excessive. This is particularly evident in the Discussion, especially paragraphs from lines 343-373 and lines 374-409. The necessary content is in the Discussion, but some of the important content 'gets lost' behind the lengthy less-important discussion points.
  8. The primary conclusion of the paper (i.e., that Arabic and PHGG fibres are likely tolerable sources of soluble fibre for patients with gastroparesis) is not stated until the last 2 short paragraphs of the Discussion. I suggest including this primary conclusion within the first paragraph of the Discussion, so that readers do not need to search the long Discussion to find the main conclusion. Starting the Discussion with a brief 'summary' paragraph could be useful - e.g., what did you do, and what was the main finding?

Specific/minor comments & typos:

  1. Intro, Line 59: "such bloating" >> "such as bloating".
  2. Intro, Line 87: "in order identify" >> "in order to identify".
  3. Methods, Line 103: "were from purchased from" >> "were purchased from".
  4. Methods, Line 112: "is shown Table 1" >> "is shown in Table 1".
  5. Table 1: consider including where each compound was sourced / purchased from. Those details were somewhat confusing and long in the text, and it would likely simplify the methods by including in the table instead.
  6. Methods, Line 140: "at pH 4 is used" >> "at pH 4 was used".
  7. Figure 1: This figure is very large, with lots of relatively small text. I am concerned about how this will translate to a formatted PDF. Can it be simplified / stream-lined, and rotated to fit in the portrait aspect of an A4 page?
  8. Methods, Line 187: Equations are typically formatted on their own line of the manuscript. I suggest doing so here. 
  9. Results, Line 261: "changes in yield stress was observed" >> "changes in yield stress were observed".
  10. Results, Line 275: "citrus pectin exhibits no significant" >> "citrus pectin exhibited no significant".
  11. Results, Line 276: "pectin exhibits a" >> "pectin exhibited a".
  12. Results, Line 283: "mg/mL shown as in Figure 6" >> mg/mL as shown in Figure 6".
  13. Results, Line 291: "compared its" >> "compared to its".
  14. Discussion, Line 307: "pyloris" >> "pylorus".
  15. Discussion, Line 315: "demonstrate this is, as PHGG" >> "demonstrate this, as PHGG".
  16. Discussion, Line 327: "this general pattern". This statement is vague. Which general pattern?
  17. Discussion: Throughout the Discussion, many sentences begin with "This...", which is ambiguous and difficult to follow what the authors are referring to. E.g., Line 329, "This is due to..."; Line 335. "This is due to..."; Line 339, "This may be due to..."; etc. 
  18. Discussion, Line 343: "in literature the rheology" >> "in literature that the rheology".
  19. Discussion, Line 437: "Our results show inulin to significantly increase viscosity in simulated digestion and over time." What is the time course? Is it comparable to the 4-hour emptying time course for gastroparesis, which the authors compare it to in the following sentence?

Author Response

Reviewer 2

The authors would like to thank the reviewer for taking the time to review our manuscript and providing positive and useful feedback. We have taken the reviewer’s comments into consideration and our responses are provided below:

  1. The title, and paper, describes the methods as "simulated digestion". I assumed that this meant computational simulations, but in fact, the authors performed chemical simulations. I suggest expanding this for clarity. E.g., in the title, "...during chemically simulated digestion...".

Reply: The changes in method description “chemically simulated digestion” has been amended throughout the manuscript as reviewer suggested.

  1. Intro, Line 57: The authors state, "Sufferers experience a range of post-prandial symptoms due to delayed gastric emptying." I am not aware of a defined cause-effect relationship between symptoms and gastric-emptying in gastroparesis. This statement should be referenced, although I ultimately suggest changing to the wording to reflect an association rather than a cause-effect. E.g., >>"...symptoms associated with delayed gastric emptying."

Reply: The statement has been modified to reflect reviewer suggestion [Line 60-61].

  1. Intro, Line 62: The statement on long-term health complications would benefit from including a reference(s). 

Reply: Reference citation has been added to the statement as suggested [Line 65-68].

  1. Figures 2&3: How are the results different for pH 2 vs 4 vs 7? They all the same / similar to me. It would be good to mention the differences, or lack thereof, in the caption. 

Reply: The similarity of rheological behaviour under both distilled water and simulated digestion is now stated in the Figure 2&3 captions.

  1. Figures 4, 5&6: Are you able to include the stats? Which of these results are statistically significant compared to the others?

Reply: Data tips indicating statistical significance have been added to Figures 4, 5&6 as suggested. A statement regarding the data tips has also been added in the figure captions.

  1. Figures 4, 5&6: Green and grey are both used twice. It is impossible to tell which bar is which in the graph. I assume it follows the order in the legend at the bottom, but that is just an assumption. Different colours should be used for clarity.

Reply: The colours have been updated for clarity as suggested.

  1. The paper is extremely well written, but it is also very long. Each of the sections are lengthy, and at some points, the length is excessive. This is particularly evident in the Discussion, especially paragraphs from lines 343-373 and lines 374-409. The necessary content is in the Discussion, but some of the important content 'gets lost' behind the lengthy less-important discussion points.

Reply: The previous section (lines 343-373) has been reviewed and edited to remove less important information [Lines376-393].

  1. The primary conclusion of the paper (i.e., that Arabic and PHGG fibres are likely tolerable sources of soluble fibre for patients with gastroparesis) is not stated until the last 2 short paragraphs of the Discussion. I suggest including this primary conclusion within the first paragraph of the Discussion, so that readers do not need to search the long Discussion to find the main conclusion. Starting the Discussion with a brief 'summary' paragraph could be useful - e.g., what did you do, and what was the main finding?

Reply: The primary discussion points have been moved to the start of the discussion as suggested. A new conclusion section has been created for readers who wish to read a quick summary of the main findings of the study and any future directions.

Specific/minor comments & typos:

  1. Intro, Line 59: "such bloating" >> "such as bloating".

Reply: Changed as suggested [Line 62]

  1. Intro, Line 87: "in order identify" >> "in order to identify".

Reply: Changed as suggested [Line 93].

  1. Methods, Line 103: "were from purchased from" >> "were purchased from".

Reply: Changed as suggested [Line 109].

  1. Methods, Line 112: "is shown Table 1" >> "is shown in Table 1".

Reply: Changed as suggested [Line 110].

  1. Table 1: consider including where each compound was sourced / purchased from. Those details were somewhat confusing and long in the text, and it would likely simplify the methods by including in the table instead.

Reply: Commercial manufacturers of test fibres have been added to Table 1.

  1. Methods, Line 140: "at pH 4 is used" >> "at pH 4 was used".

Reply: Changed as suggested [Line 139].

  1. Figure 1: This figure is very large, with lots of relatively small text. I am concerned about how this will translate to a formatted PDF. Can it be simplified / stream-lined, and rotated to fit in the portrait aspect of an A4 page?

Reply: The flowchart has been modified into an A4 portrait layout. The font size and has been increased and the clarity of presentation improved as suggested.

  1. Methods, Line 187: Equations are typically formatted on their own line of the manuscript. I suggest doing so here. 

Reply: The two main rheological equations are now displayed in separate lines for clarity [Lines 186 and 196].

  1. Results, Line 261: "changes in yield stress was observed" >> "changes in yield stress were observed".

Reply: Changed as suggested [Line 269].

  1. Results, Line 275: "citrus pectin exhibits no significant" >> "citrus pectin exhibited no significant".

Reply: Changed as suggested [Lines 287-288].

  1. Results, Line 276: "pectin exhibits a" >> "pectin exhibited a".

Reply: Changed as suggested [Lines 288-289].

  1. Results, Line 283: "mg/mL shown as in Figure 6" >> mg/mL as shown in Figure 6".

Reply: Changed as suggested [Line 299].

  1. Results, Line 291: "compared its" >> "compared to its".

Reply: Changed as suggested [Line 307].

  1. Discussion, Line 307: "pyloris" >> "pylorus".

Reply: Grammatical error changed as suggested [line 331].

  1. Discussion, Line 315: "demonstrate this is, as PHGG" >> "demonstrate this, as PHGG".

Reply: Changed as suggested [Line 340].

  1. Discussion, Line 327: "this general pattern". This statement is vague. Which general pattern?

Reply: Changed as suggested [353 – 355].

  1. Discussion: Throughout the Discussion, many sentences begin with "This...", which is ambiguous and difficult to follow what the authors are referring to. E.g., Line 329, "This is due to..."; Line 335. "This is due to..."; Line 339, "This may be due to..."; etc.

Reply: “This may” and “This is” grammatical statements have been modified as suggested to remove any ambiguity within the statement.

  1. Discussion, Line 343: "in literature the rheology" >> "in literature that the rheology".

Reply: Changed as suggested [Line 369].

  1. Discussion, Line 437: "Our results show inulin to significantly increase viscosity in simulated digestion and over time." What is the time course? Is it comparable to the 4-hour emptying time course for gastroparesis, which the authors compare it to in the following sentence?

Reply: The statement has been modified to properly reflect the rheological and clinical implications [Lines 401 – 407]